# Long-lasting insecticidal nets (LLINs) use among household members for protection against mosquito bite in Mogadishu districts

**Ahmed Aweis**[1]*, **Abdinur A. Salad**[1], **Fathi A. Araye**[2], **Abdifatah M. Ahmed**[3], **Osman A. Wehlie**[1], **Ali Abdirahman Osman**[4], **Isaiah Gumbe Akuku**[5]

**1** Benadir University, Mogadishu, Somalia, **2** Jazeera University Hospital, Mogadishu, Somalia, **3** Somali International University, Mogadishu, Somalia, **4** Ministry of Health, Federal Government of Somalia, Mogadishu, Somalia, **5** Institute of Tropical and Infectious Diseases, University of Nairobi, Nairobi, Kenya

* draweys87@gmail.com

**Data Availability Statement:** All data are in the supporting information file.

## Abstract

Understanding long-lasting insecticidal nets (LLINs) utilization is important in monitoring and quantifying the impact of past and current prevention and control efforts of malaria. A cross-sectional study was carried out on a sample of 409 households in Mogadishu, to estimate the LLIN use and assess barriers to its utilization. A standardized questionnaire was used to collect data on demographics, malaria-related knowledge, and the use of preventive measures. LLINs use was assessed using multivariable generalized estimating equations with adjustment for clustering of study participants within the same household. Out of 409 households only 155 (37.9%) owned LLINs. Out of 237 owned LLINs, 199 (84.0%) were used. Median household size being 6.0 (3.0), intra-household net accessibility was low, with one net (42.6%) frequent. Most nets were from mass distribution (55.7%) and obtained '12 months ago'. Un-partnered respondents (unadjusted odds ratio [OR] 0.34, 95% CI 0.14, 0.82; p = 0.017) compared with partnered (married) respondents, large-sized household (adjusted OR 0.83, 96% CI 0.74–0.94; p = 0.002). There was marginal evidence of a greater odds of LLIN utilization among respondents knowledgeable of the correct cause of malaria, that is, mosquito bites (AOR 3.19, 95% CI 0.77, 13.2; p = 0.11) but was not statistically significant. Among households owning nets, most of the LLINs were hung the night prior to the survey (7.9% versus 98%) and was associated with greater marginal odds of utilization (p<0.001). Ownership of LLINs is insufficient in Mogadishu districts affecting household-level access and utilization. If this is not checked, this could weaken the progress made on malaria control efforts. LLIN utilization was modest and largely driven by recently acquired nets showing a desire to utilize them despite low coverage. These imply that mass and facility-based distribution, and awareness campaigns will remain relevant, but efforts for willingness-to-pay for LLINs should be strengthened to sustain coverage and replacements of worn-out nets.

**Funding:** The authors received no specific funding for this work.

**Competing interests:** The authors have declared that no competing interests exist.

## Introduction

Malaria remains a major public health problem in Somalia and nearly 90% of Somalia's population is at risk of malaria according to previous estimates [1] Plasmodium falciparum is responsible for about 90% of cases of malaria in Somalia. The epidemiological feature of malaria in Somalia is divided into hypoendemic (North), mesoendemic to hypoendemic in the Centre and South and hyper-endemic in the riverine areas of the Juba and Shabelle rivers [2]. There is continued endemicity of malaria in most parts of the Central and Southern regions and some areas in the north with other areas being prone to epidemics. In 2015 more than 103,450 malaria cases were recorded and treated [3].

Published malariometric surveys in Mogadishu districts (Benadir region) have shown prevalence as high as 50% [4] (Malaria Indicator Survey 2014) and estimates between 0% and 52% in the south. However, a dramatic reduction in malaria cases and death was seen in 2021 compared to previous years due to the malaria control program in the country [5].

Because of the high numbers of morbidities and mortalities, it was a necessity to organize efforts to tackle this issue. One of the major recommendations of WHO in terms of intervention is the usage of long-lasting insecticide-treated nets [6]. Long-lasting insecticide-treated nets (LLINs) functions as a physical and chemical barrier and are now accepted as the most cost-effective nets for malaria prevention because the protection they offer is twice as much as that provided by untreated nets. LLINs do not require re-treatment [7].

LLINs are distributed in Mogadishu through the Ministry of Health (MOH) and partners. Because of the malaria burden, the GFATM has continued to do a major role in the distribution of LLINs to reach 90% coverage in targeted areas [8, 9]. Since 2006, strategies and interventions for malaria vector control have been reported in Somalia malaria strategic plans, mainly focused on free mass distribution and utilization of LLIN as well as Indoor Residual Spray (IRS) to specific malaria foci, primarily in Somalia's northern regions [9, 10]. Within the framework of the Somalia National Malaria Strategic plan for 2017–2020, the main vector control activity implemented in the country is LLIN distributions [10]. This involves community level free mass distribution of LLINs, and household to household distribution to provide universal coverage with LLINs across the country. According to UNICEF Somalia, from 2015 to 2017, about 2.3 million LLINs were distributed and 4.6 million households benefitted [3]. In 2017, UNICEF Somalia with GFATM partnership undertook mass distribution campaigns of about 1.2 million LLINs through MOH and local partners [3]. The goal of universal coverage is to ensure that at least 90% of households own an LLIN, and at least 80% use at least one LLINs per 2 persons. With a lot of money, manpower, and other recourses allocated to the universal coverage of LLINs. The benefit of this effort comes with sustainable and continuous usage and therefore the interplay with the use and non-use of nets.

According to the 2013 estimates from the First Malaria Indicator Survey in Somalia [4], 19.1% of households have at least one LLIN and this is substantially lower than the 80% level recommended by WHO to realize substantial malaria transmission decrease. Merely 23.0% of pregnant women and 22.5% of children under five years of age utilize an LLIN to prevent malaria infection. With the increased distribution of LLINs, there is a need for current estimates of LLIN use.

The ownership and use of LLINs have provided promising results across sub-Saharan Africa to reduce clinical episodes of malaria and all-cause child mortality [11, 12]. The distribution of LLINs particularly through universal coverage campaigns has been an extensively adopted method and several sub-Saharan Africa countries have witnessed substantial LLIN distribution activities in the past few years [13, 14].

Coverage only does not help in reducing morbidity and mortality but effective usage of the LLIN coupled with it can give the desired results. There are many studies worldwide that

found a huge discrepancy between ownership versus the use of ITNs. Studies quantified this difference as 95% vs 59% (Kenya), 70% vs 53.1% (Nigeria), and 90% vs 77% (Tanzania) indicating household ownership has not reflected the utilization of the bed net [15].

To meet the Sustainable Development Goals (SDG) and the Roll Back Malaria targets, it is crucial to determine the real usage levels and to take timely corrective actions. Thus, the objective of the study was to determine the level of LLIN use and identify factors (barriers) independently associated with their use and non-use at the household, among the residents of Mogadishu districts, and to provide evidence for advancing public health strategies for preventing transmission and controlling malaria vector, leverage political commitment, influence policy, and funding. Routine distribution remains as per the Somalia national LLIN policy guidelines focusing on pregnant women and newborns [9].

There have been many LLIN distribution campaigns. National and international organizations have distributed LLINs in all districts of Mogadishu in the years 2019 and 2020.

## Materials and methods

### Study settings

The research was conducted in Mogadishu a home of 1,650,227 making 13.4% of the total Somalia population according UNFPA 2014 data [16]. The Somalia Health and Demographic Survey (SHDS) report reveal a mean household size of 6.2 in Somalia and 6.6 urban households [17]. The 2014 malaria indicator survey reported an estimate of 4.2 in the Benadir region [4].

### Study design

This was a cross-sectional household study conducted using a stratified two-stage cluster survey design.

### Study population

The target population was all household members in Mogadishu districts. The long-lasting insecticide-treated bed nets (LLINs) were distributed in 2019/2020.

### Sample size

According to the 2014 Somalia Malaria Indicator Survey [4], 20.1% of household members slept under an LLIN the night prior to the survey. Assuming that Mogadishu districts are less heterogeneous and taking a design effect of 2.0, a sample size of 494 households to achieve a precision of 5% around the 20.1%, with 95% confidence intervals was calculated. This sample size is computed using the single population proportion formula for cross-sectional studies/surveys below [18].

$$n = DEFT\left(\frac{Z_{1-\frac{\alpha}{2}}^2 P(1-P)}{d^2}\right) = 2.0\left(\frac{1.96^2 * 0.201 * (1 - 0.201)}{0.05^2}\right) = 494$$

Where: n = sample size of the members of the household; DEFT = design effect; P = proportion of members of the households who slept under LLIN the night prior to the 2014 Somalia Malaria Indicator Survey; d = degree of accuracy or precision required (the sampling error) of 5% or 0.05; $Z_{1-\alpha/2}$ = is the standard normal $Z$ value that corresponds to a cumulative probability of 1 –α/2. Accounting for the non-response rate, a further 10% was taken to adjust it to 544 households. Due to COVID-19 restrictions, we reached 409 households with a

total of 2442 members and a subset of 942 household members within 155 households owning nets.

## Sampling frame and sampling procedure

The sampling frame was a complete list of geographic information system spatially constructed (digitized) dwelling structures in the Mogadishu districts described in the SHDS 2020 and UNFPA [17]. In the SHDS 2020, the urban areas have 50–49 dwelling structures within each of the statistical units (enumeration areas–EAs) as shown in Omar et al. [19].

In the first sampling stage, 11 EAs were selected in each district by random sampling by probability proportional to size (PPS) of digitized dwelling structures. In the second stage, systematic random sampling was used to 8 households in each EA.

We envisaged a two-stage stratified cluster sampling design, however, due to the COVID-19 pandemic, we adjusted to a three-stage cluster design. In the first sampling stage, we selected seven districts randomly out of a total of 18 Mogadishu districts. We then sampled 11 enumeration areas out of each of the seven districts in the second stage by probability proportional to the estimated size (PPS) of digitized dwelling structures described in the SHDS 2020. In the third stage, we systematically sampled eight households selected from each EA

Households were selected by dividing the digitized EA into quadrants, a central point was chosen within each of the assigned quadrants, an ink pen or bottle was spun to select a direction to proceed, and choosing the first/initial household in this direction to constitute the starting point and approaching the adjacent households systematically for an interview. In case eligible households are more than one in a house, one eligible household was randomly chosen. A household was defined as "a group of people living together in the same housing (dwelling) unit, who eat food prepared in the same cooking pot, and who recognized the same person as the head of the household" consistent with definitions used in previous surveys in Somalia and elsewhere [17].

Data was collected on all household members who slept in the household ("sharing a common cooking pot") the night before the survey.

## Data collection and survey questionnaire

Three research teams each made up of a team leader (supervisor) and three interviewers administered a face-to-face survey questionnaire over five weeks, between 1st November 2020 and 31st March 2021. The research teams used a structured questionnaire through verbal interviews and direct observation.

The questionnaires adapted from the Roll Back Malaria Monitoring and Evaluation Reference Group (RB-MERG) Malaria Indicator Survey [32]. The questionnaires were translated into Somali language and was pre-tested before the main collection of data to check if the questions' wording is appropriate and verification of the translated texts and skip logic. In the 2020 SHDS, the Benadir region was considered fully urban and therefore no need to stratify Mogadishu districts into rural and urban.

We asked the household heads/respondents their age (years), gender, education, marital status, average Monthly Income (USD), distance to the nearest health facility/ hospital in kilometers (Km), the number of people who slept last night, the number of sleeping places in household, the number of sleeping spaces, whether the interior walls of dwelling sprayed against mosquitoes and the sleeping materials in household. We also asked the respondents the main cause of malaria, the main source of information about use of bed nets, the main method used to protect against malaria, whether household member(s) suffered malaria in the past 12 months, and the particular members that suffered malaria last 12 months. Additionally, we recorded the reasons not owning

nets, the physical appearance, source of the nets and utilization LLIN of a particular net (net-level) by the household members the night prior to the survey. We also asked the respondents the categories of people that use the nets when there is only one net in the household. This was particularly useful to characterize the groups of at risk population, especially the infants and others generally below the age of five as well as the pregnant women who are susceptible to malaria in pregnancy and how this influences net ownership and utilization.

## Data management and statistical analysis

Data was transferred regularly to a private secure server at Somali research and Development Institute (SORDI)/Benadir University. The de-identified data was then imported into an Excel$^{TM}$ spreadsheet and counterchecked to exclude or harmonize any inconsistencies that may occur. The statistical analyses were carried out in R software version 4.2.1. [20].

For the study participants' sociodemographic and household characteristics, descriptive statistics were calculated such as medians appropriately and the categorical variables as frequencies, percentages/proportions.

Factors independently associated with use (net-level use or non-use) of a particular LLIN the previous night was the dependent variable. The LLIN-level. The LLIN-level use and non-use of was assessed using both univariable and multivariable generalized estimating equations (GEE) with adjustment for clustering of study net-level The GEE models were constructed usingan exchangeable correlation structure and a binomial logit link to determine independent predictors of and/or barriers to net-level LLIN use and account for clustering of survey of nets at household level (i.e., the within-household net use variability) [21].

## Ethical considerations

This study was reviewed and approved by Somali Medical Association (SMA).

Participation in this household study was completely voluntary, and no incentive was given for participation. Verbal informed consent was also sought from the study respondents for selected households.

## Results

### Household- and household head's individual level characteristics

A total of 409 households responded to this survey of which 155 (37.9%) owned LLINs. The respondents heads were mostly female (87.3%) (Table 1). The median age of respondents in years was 30.0 (interquartile range [IQR] = 15.0). About 25% reported having attained primary educational level, 12% attained secondary or tertiary level of education and 63.1% were illiterate. Majority of the respondents were married (73.3%). The median of average income in United States Dollar (USD) was 100.0 (90.0) of which the majority (49.9%) were in the category of USD 60–100. Households without mosquito nets (LLINs) had higher average income than those owning the nets (median greatest dimension, 100.0 vs 90.0; p<0.001). Household owning nets were further away from their nearest health facility/hospital than households not owning the nets (median greatest dimension, 2.0 Km vs 1.0 Km; p<0.001).

The median number of people who slept in the house the night prior to the survey was 5.0 (3.0). The main sleeping place was a two–bed roomed household among 335 (81.9%) respondents. Increase in number of sleeping spaces was associated with owning a mosquito net (median greatest dimension, 3.0 vs 2.0; p = 0.001). Indoor residual spraying was associated with owning a net (27.1% vs 15.7%; p<0.001). Additional individual- and household-level characteristics stratified by LLIN ownership are presented in Table 1.

**Table 1. Household head's individual- and household-level characteristics stratified by ownership of LLIN among households surveyed, Mogadishu–Somalia, 2021.**

| Characteristics | Level | Total | Household own mosquito net | | |
|---|---|---|---|---|---|
| | | | **Yes** | **No** | **p-value[1]** |
| Total N (%) | | 409 | 155 (37.9) | 254 (62.1) | |
| Age (years) | Median (IQR) | 30.0 (15.0) | 30.0 (15.0) | 30.0 (15.0) | 0.508 |
| Age group (years) | <25, n (%) | 95 (23.2) | 36 (23.2) | 59 (23.2) | 0.829 |
| | 25–30, n (%) | 134 (32.8) | 47 (30.3) | 87 (34.3) | |
| | 31–40, n (%) | 90 (22.0) | 35 (22.6) | 55 (21.7) | |
| | 41+, n (%) | 90 (22.0) | 37 (23.9) | 53 (20.9) | |
| Gender | Male, n (%) | 52 (12.7) | 22 (14.2) | 30 (11.8) | 0.583 |
| | Female, n (%) | 357 (87.3) | 133 (85.8) | 224 (88.2) | |
| Education | Illiterate, n (%) | 258 (63.1) | 101 (65.2) | 157 (61.8) | 0.900 |
| | Primary, n (%) | 102 (24.9) | 37 (23.9) | 65 (25.6) | |
| | Secondary, n (%) | 36 (8.8) | 12 (7.7) | 24 (9.4) | |
| | Tertiary, n (%) | 13 (3.2) | 5 (3.2) | 8 (3.1) | |
| Marital status | Married, n (%) | 300 (73.3) | 124 (80.0) | 176 (69.3) | 0.059 |
| | Single, n (%) | 35 (8.6) | 10 (6.5) | 25 (9.8) | |
| | Widowed/ Divorced/ Separated, n (%) | 74 (18.1) | 21 (13.5) | 53 (20.9) | |
| Average Monthly Income (USD[2]) | Median (IQR) | 100.0 (90.0) | 90.0 (40.0) | 100.0 (83.8) | **<0.001** |
| | <60, n (%) | 74 (18.1) | 37 (23.9) | 37 (14.6) | **0.014** |
| | 60–100, n (%) | 204 (49.9) | 81 (52.3) | 123 (48.4) | |
| | 101–150, n (%) | 67 (16.4) | 21 (13.5) | 46 (18.1) | |
| | 150+, n (%) | 64 (15.6) | 16 (10.3) | 48 (18.9) | |
| Distance to the nearest health facility/ hospital (Km) | Median (IQR) | 1.0 (1.0) | 2.0 (2.0) | 1.0 (0.0) | **<0.001** |
| Number of people who slept last night | Median (IQR) | 5.0 (3.0) | 6.0 (3.0) | 5.0 (3.0) | 0.518 |
| Sleeping places in household | Open room, n (%) | 59 (14.4) | 28 (18.1) | 31 (12.2) | 0.144 |
| | 2–Bed room, n (%) | 335 (81.9) | 119 (76.8) | 216 (85.0) | |
| | Open room and 2–Bed room, n (%) | 13 (3.2) | 7 (4.5) | 6 (2.4) | |
| | Other[3], n (%) | 2 (0.5) | 1 (0.6) | 1 (0.4) | |
| Number of sleeping spaces | Median (IQR) | 2.0 (3.0) | 3.0 (3.0) | 2.0 (2.0) | **0.001** |
| Interior walls of dwelling sprayed against mosquitoes | No, n (%) | 323 (79.0) | 113 (72.9) | 210 (82.7) | **0.005** |
| | Yes, n (%) | 82 (20.0) | 42 (27.1) | 40 (15.7) | |
| | Do not know, n (%) | 4 (1.0) | 0 (0.0) | 4 (1.6) | |
| Sleeping materials in household | Mats, n (%) | 54 (13.2) | 30 (19.4) | 24 (9.4) | **<0.001** |
| | Mattress, n (%) | 176 (43.0) | 96 (61.9) | 80 (31.5) | |
| | Beds, n (%) | 178 (43.5) | 28 (18.1) | 150 (59.1) | |
| | Other[4], n (%) | 1 (0.2) | 1 (0.6) | 1 (0.0) | |

[1] p-value from Chi-square test or Fisher's exact test for categorical variables and unpaired two-sample Wilcoxon/Mann-Whitney U test for continuous variables, two-sided; bold p-values indicate statistical significance ($p<0.05$).

[2] USD: United States Dollar.

[3] 5-bed room.

[4] Traditional bed

## Knowledge of causes, preventive/protective measures and information about LLINs as reported by the head of the household

Most respondents (44.4%) knew about the cause of malaria (Table 2), and this was not significantly associated with owning a net (83.9 vs 81.9; p = 0.7131). The main source of information about use of bed nets was community-based volunteer/leader (44.0%). Household member

**Table 2. Reported knowledge of causes and preventive measures and information about bed nets use among households as reported by the household heads in the Mogadishu districts, Somalia, 2021.**

| Factor | Levels | Total (%)[1] | Household own mosquito net | | |
|---|---|---|---|---|---|
| | | | Yes (%) | No (%) | p-value[2] |
| Main cause of malaria | Mosquito bites/ | 340 (44.4) | 127 (42.6) | 213 (45.6) | **0.005** |
| | Eating other dirty food | 23 (3.0) | 14 (4.7) | 9 (1.9) | |
| | Drinking dirty water | 95 (12.4) | 43 (14.4) | 52 (11.1) | |
| | Getting soaked with rain | 66 (8.6) | 36 (12.1) | 30 (6.4) | |
| | Cold or changing weather | 46 (6.0) | 23 (7.7) | 23 (4.9) | |
| | Dirty surrounding | 170 (22.2) | 44 (14.8) | 126 (27.0) | |
| | Witchcraft | 6 (0.8) | 4 (1.3) | 2 (0.4) | |
| | Other cause(s)[3] | 8 (1.0) | 6 (2.0) | 2 (0.4) | |
| | Do not know | 11 (1.4) | 1 (0.3) | 10 (2.1) | |
| | *Mosquito bites/also mentioned other cause* | 340 (83.1) | 127 (81.9) | 213 (83.9) | 0.713 |
| | *All other incorrect* | 69 (16.9) | 213 (83.9) | 41 (16.1) | |
| Main cause of malaria (mosquito bites vs. incorrect) | *Mosquito bites* | 340 (83.1) | 127 (81.9) | 213 (83.9) | 0.713 |
| | *All other incorrect* | 69 (16.9) | 213 (83.9) | 41 (16.1) | |
| Main source of information about use of bed nets | Radio/Television | 105 (20.6) | 43 (20.0) | 62 (21.1) | **0.012** |
| | Health facility | 118 (23.2) | 64 (29.8) | 54 (18.4) | |
| | Community based volunteer/leader | 224 (44.0) | 82 (38.1) | 142 (48.3) | |
| | Neighbour/ Relative | 31 (6.1) | 15 (7.0) | 16 (5.4) | |
| | Social media | 8 (1.6) | 3 (1.4) | 5 (1.7) | |
| | No information | 18 (3.5) | 4 (1.9) | 14 (4.8) | |
| | Other[4] | 5 (1.0) | 4 (1.9) | 1 (0.3) | |
| Main method used to protect against malaria | Sleep under a bednet | 169 (27.8) | 97 (38.6) | 72 (20.2) | **<0.001** |
| | Sleep under an insecticide-treated bednet | 89 (14.7) | 64 (25.5) | 25 (7.0) | |
| | Use mosquito repellent | 40 (6.6) | 19 (7.6) | 21 (5.9) | |
| | Take preventive medication | 22 (3.6) | 14 (5.6) | 8 (2.2) | |
| | Spray house with insecticide | 221 (36.4) | 35 (13.9) | 186 (52.2) | |
| | Keep house surroundings clean | 24 (4.0) | 10 (4.0) | 14 (3.9) | |
| | Other[5] | 17 (2.8) | 10 (4.0) | 15 (4.2) | |
| | Do not know | 25 (4.1) | 2 (0.8) | 15 (4.2) | |
| Household member suffered malaria | Yes | 257 (62.8) | 113 (72.9) | 144 (56.7) | **0.001** |
| | No/ Do not know | 152 (37.2) | 42 (27.1) | 110 (43.3) | |
| Members suffered malaria last 12 months | Children under 5 years | 96 (27.7) | 39 (26.4) | 57 (28.8) | 0.770 |
| | Children over 5 years | 93 (26.9) | 42 (28.4) | 51 (25.8) | |
| | Pregnant women | 27 (7.8) | 13 (8.8) | 14 (7.1) | |
| | Adults | 104 (30.1) | 41 (27.7) | 63 (31.8) | |
| | Elderly | 25 (7.2) | 13 (8.8) | 12 (6.1) | |
| | Other[6] | 1 (0.3) | 0 (0.0) | 1 (0.5) | |

[1] Percentage total may exceed the 409 due to multiple responses

[2] p-value from Chi-square test or Fisher's exact test for categorical variables, two-sided; bold p-values indicate statistical significance (p<0.05).

[3] "Duumo", wealthy bed bugs cause this person to have it, Trees, Green area.

[4] I wear a mosquito net when I hear it, In my brain and minds.

[5] Drink "carmo" to clear my stomach, light trees, eat fish (black fish) from river and lemons to prevent it, ran out of the river, Smoke, burn cultural tree, electrical killer, fuel oil, Smoked meat.

[6] Maya–don't know.

having suffered malaria was associated with owning a mosquito net (72.9 vs 56.7; p = 0.001). On the main method used to protect against malaria, sleeping under a bednet (27.8%) was the most cited method.

### Factors associated with net ownership

In the univariable analysis, we found that household heads/respondents who were widowed/ divorced or separated and an increase in the number of people who slept the previous night were associated with lower odds of net ownership. Additionally, we did not find a clear significant association between net ownership and income when we considered the categorical levels and continuous income as a unit increase in income–some categories portrayed a significant decrease in odds of net ownership compared to the lowest band (< USD 60) while on the metric income indicated a negligible increase in odds with a unit increase in income (mixed results). However, an additional sleeping space, a kilometer (Km) increase in distance to the nearest health facility or hospital, households with interior walls of the dwelling sprayed against mosquitoes at any time in the past 12 months, households that reported that member (s) had suffered malaria in the last 12 months, and households with mats/mattresses as sleeping materials compared to beds was associated with a decrease in net ownership.

In the multivariable model, distance to the nearest health facility or hospital (adjusted odds ratio [AOR] 1.01, 95% CI: 1.00, 1.02; p = 0.046), dwelling with interior walls sprayed against mosquitoes at any time in the past 12 months (AOR 2.88, 95% CI: 1.60, 5.29; p<0.001) and sleeping materials in mats/mattresses compared to beds (AOR 5.48, 95% CI: 3.22, 9.62; p<0.001) was significantly associated with net ownership. Table 3 below summarizes both the univariable and multivariable models in the net ownership.

### Reason households do not own a LLIN

The most common reason stated for not owning an LLIN was that it is too expensive (42.0%) while the least cited was "change my sleeping space too often" and "it does not protect against mosquito bites/insects" each at 1.6% of the total responses. Of those who stated that "bed nets not available" and that "it is too hot under the net", 32.1% (34/106) and 85.7% (12/14) had had household member(s) suffer malaria in the past 12 months, respectively. Other reasons are summarized in Table 4 below.

### Barriers of utilization of LLIN

**Utilization, physical appearance and variations of LLIN by source.** The median nets per household was 2 (1.0, 2.0) nets in this study. Out of 237 owned LLINs in the 155 surveyed net-owned households, 199 (84.0%) slept under the night prior to the survey. There was lower intra-household accessibility of the nets, and mostly, there were 101 (42.6%) households with one net, 86 (36.3%) with two nets and 21 (8.9%) three nets, despite there being about six people who slept in the household the previous night (Median, 6.0 (IQR, 3.0), therefore universal coverage of the nets in each household was not attained.

Most of the nets were from mass distribution (55.7%) and market/retail shop (19.5%), out of which 78.8% [108/137] and 93.8% [45/48], respectively, were slept under the night prior to the survey. The median time since the nets were obtained was '12 months ago'. The nets were more likely to be slept under when they were two or more in the household (60.8% vs 39.2%, p = 0.024; median greatest dimension 2 nets vs 1 net, p = <0.001). In cases where there was only one net, the net was majorly used by children under five years old (37.3%) followed by children under five (not infants) and pregnant women, however, the difference in frequency by use of the net the previous night was not statistically significant (p = 0.190). The nets

**Table 3. Logistic regression model results of factors associated with net ownership.**

| | | | Household own mosquito net | | | | |
|---|---|---|---|---|---|---|---|
| Variable | | Univariable Model | | | Multivariable Model | | |
| | N | OR[1] | 95% CI[1] | p-value | OR[1] | 95% CI[1] | p-value |
| Age group (years) | 409 | 1.01 | 0.99, 1.02 | 0.43 | | | |
| <25 | | — | — | | | | |
| 25–30 | | 0.89 | 0.51, 1.53 | 0.66 | | | |
| 31–40 | | 1.04 | 0.58, 1.89 | 0.89 | | | |
| 41+ | | 1.14 | 0.63, 2.07 | 0.65 | | | |
| Sex of the respondent | 409 | | | | | | |
| Male | | — | — | | | | |
| Female | | 0.81 | 0.45, 1.48 | 0.48 | | | |
| Education | 409 | | | | | | |
| Illiterate | | — | — | | | | |
| Primary | | 0.88 | 0.55, 1.42 | 0.61 | | | |
| Secondary | | 0.78 | 0.36, 1.60 | 0.50 | | | |
| Tertiary | | 0.97 | 0.29, 2.99 | 0.96 | | | |
| Marital status | 409 | | | | | | |
| Married | | — | — | | — | — | |
| Single | | 0.57 | 0.25, 1.19 | 0.15 | 0.48 | 0.20, 1.11 | 0.094 |
| Widowed/Divorced/Separated | | 0.56 | 0.32, 0.97 | **0.042** | 0.57 | 0.30, 1.05 | 0.077 |
| Average Monthly Income (United States Dollar) | 409 | 1.00 | 0.99, 1.00 | **0.008** | 1.00 | 1.00, 1.00 | 0.31 |
| Average Monthly Income—Quartiles (United States Dollar)) | 409 | | | | | | |
| <60 | | — | — | | | | |
| 60–100 | | 0.66 | 0.38, 1.13 | 0.13 | | | |
| 101–150 | | 0.46 | 0.23, 0.90 | **0.026** | | | |
| 150+ | | 0.33 | 0.16, 0.68 | **0.003** | | | |
| Distance to the nearest health facility or hospital (Km) | 409 | 1.01 | 1.01, 1.02 | **0.001** | 1.01 | 1.00, 1.02 | **0.046** |
| Number of people per sleeping space | 409 | 1.01 | 0.96, 1.05 | 0.71 | 0.99 | 0.94, 1.04 | 0.74 |
| Sleeping places in the household | 409 | | | | | | |
| Open room | | — | — | | | | |
| 2-Bed room | | 0.61 | 0.35, 1.07 | 0.082 | | | |
| Open room and 2-Bed room | | 1.29 | 0.38, 4.46 | 0.68 | | | |
| Other (specify) | | 1.11 | 0.04, 28.9 | 0.94 | | | |
| Sleeping spaces | 409 | 1.12 | 1.03, 1.22 | **0.008** | 1.10 | 1.00, 1.20 | 0.054 |
| Number of people who slept last night | 334 | 0.86 | 0.75, 0.99 | **0.037** | | | |
| Interior walls of dwelling sprayed against mosquitoes at any time in the past 12 months | 409 | 1.99 | 1.22, 3.25 | **0.006** | 2.88 | 1.60, 5.29 | **<0.001** |
| Household member(s) suffered malaria in the last 12 months | 409 | 2.06 | 1.34, 3.19 | **0.001** | 1.36 | 0.82, 2.25 | 0.23 |
| Category of household members suffered malaria last 12 months | 256 | | | | | | |
| Adults | | — | — | | | | |
| Children over 5 | | 1.19 | 0.61, 2.33 | 0.62 | | | |
| Children under 5 years | | 0.90 | 0.48, 1.72 | 0.76 | | | |
| Elderly | | 1.65 | 0.40, 7.21 | 0.48 | | | |
| Pregnant women | | 1.10 | 0.29, 4.02 | 0.88 | | | |
| Other (specify) | | 0.00 | | 0.99 | | | |
| Sleeping materials in household | 409 | | | | | | |
| Beds | | — | — | | — | — | |

(*Continued*)

**Table 3.** (Continued)

| Variable | | Household own mosquito net | | | | | |
|---|---|---|---|---|---|---|---|
| | | Univariable Model | | | Multivariable Model | | |
| | N | OR[1] | 95% CI[1] | p-value | OR[1] | 95% CI[1] | p-value |
| *Mats/Mattress* | | 6.27 | 3.94, 10.2 | <0.001 | 5.48 | 3.22, 9.62 | <0.001 |

[1] OR = Odds Ratio

CI = Confidence Interval

—denotes reference level.

majorly had holes (40.9%), looked old (24.5%) of looked dirty /changed colour (11.4%) among other physical appearances as shown in Table 5 below.

Table 6 below shows that 134 (89.3%) of the respondents stated they washed the LLINs. Most of the net washing was done monthly (66.4%).

**Univariable and multivariable analysis for factors independently associated with LLIN utilization and non-utilization.** The results of the multivariable analysis for factors independently associated with LLIN utilization are shown in Table 7.

The multivariable adjusted analyses were limited to households owning at least one LLIN. Households with LLINs demonstrated marginal evidence of a lower odds of LLIN utilization among unpartnered household (single/widowed/divorced/separated) respondents (OR 0.34, 95% CI 0.14, 0.82; p = 0.017) compared with partnered (married) respondents in the univariable model only. We observed that of LLIN utilization reduced with increase (large) size of the household. Specifically, a one-unit increase in the number of people who slept the night prior to the survey was associated with lower odds of utilizing the LLIN (AOR 0.83, 96% CI 0.74–0.94; p = 0.002). Compared to households that obtained LLIN not more than 12 months ago, we did not find a difference in odds of LLIN in household that obtained more than 12month

**Table 4. Reported reasons for the households not owning an LLIN, Mogadishu–Somalia, 2021.**

| Reason | | Total (%)[1] | Household member(s) suffered from malaria in the last 12 months | |
|---|---|---|---|---|
| | | | Yes (%) | No (%) |
| It is too expensive | | 154 (42.0) | 94 (50.8) | 60 (33.0) |
| Bed nets not available | | 106 (28.9) | 34 (18.4) | 72 (39.6) |
| Not enough nets for everyone in the house in the house | | 28 (7.6) | 19 (10.3) | 9 (4.9) |
| No mosquitoes around | | 28 (7.6) | 2 (1.1) | 26 (14.3) |
| Do not know where to buy one | | 16 (4.4) | 10 (5.4) | 6 (3.3) |
| It is too hot under the net | | 14 (3.8) | 12 (6.5) | 2 (1.1) |
| Change my sleeping space too often | | 6 (1.6) | 3 (1.6) | 3 (1.6) |
| It does not protect against mosquito bites/insects | | 6 (1.6) | 5 (2.7) | 1 (0.5) |
| Other | It's no more; It's sold; My brother gave out; had not sought; Don't need; Husband died early. | 9 (2.5) | 6 (3.2) | 1 (1.6) |

[1] Percentage total may exceed 254 for households not owning the LLINs due to multiple responses

**Table 5. Physical appearance, variations by source and patterns of utilization LLIN amongst household members the night prior to the survey in Mogadishu districts, Somalia, 2021.**

| Variable | Level | Total (%)[1] | Net used previous night | | |
|---|---|---|---|---|---|
| | | | Yes (%) | No (%) | p-value[2] |
| Total N (%) | | 237 | 199 (84.0) | 38 (16.0) | – |
| Source of nets | Mass distribution | 137 (55.7) | 108 (51.9) | 29 (76.3) | 0.137 |
| | ANC | 17 (6.9) | 16 (7.7) | 1 (2.6) | |
| | Market/Retail shop | 48 (19.5) | 45 (21.6) | 3 (7.9) | |
| | Health facility | 28 (11.4) | 24 (11.5) | 4 (10.5) | |
| | Pharmacy | 6 (2.4) | 6 (2.9) | 0 (0.0) | |
| | Friend/relative | 10 (4.1) | 9 (4.3) | 1 (2.6) | |
| Lifespan of the LLINs (obtained net) | Median (Q1-Q3)[3] | 12 (12–24) | 12 (12–24) | 12 (12–24) | – |
| | <12 months ago | 53 (22.4) | 47 (23.6) | 6 (15.8) | **0.005** |
| | 12 months ago | 101 (42.6) | 86 (43.2) | 15 (39.5) | |
| | 13 to <24 months ago | 46 (19.4) | 42 (21.1) | 4 (10.5) | |
| | 25 to 48 months ago | 37 (15.6) | 24 (12.1) | 13 (34.2) | |
| Number of nets available (continuous) | Median (Q1-Q3) | 2 (1.0, 2.0) | 2 (1.0, 2.0) | 1 (1.0, 2.0) | **<0.001** |
| Number of nets available (groups) | One | 101 (42.6) | 78 (39.2) | 23 (60.5) | **0.012** |
| | Two | 86 (36.3) | 76 (38.2) | 10 (26.3) | |
| | Three | 21 (8.9) | 21 (10.6) | 0 (0.0) | |
| | Four | 4 (1.7) | 4 (2.0) | 0 (0.0) | |
| | Five | 15 (6.3) | 10 (5.0) | 5 (13.2) | |
| | Ten | 10 (4.2) | 10 (5.0) | 0 (0.0) | |
| | One | 101 (42.6) | 78 (39.2) | 23 (60.5) | **0.024** |
| | 2 or more | 136 (57.4) | 121 (60.8) | 15 (39.5) | |
| Number of nets available (one vs. ≥2) | One | 101 (42.6) | 78 (39.2) | 23 (60.5) | **0.024** |
| | 2 or more | 136 (57.4) | 121 (60.8) | 15 (39.5) | |
| People using net if there is only one net available in the household (n = 100) | Infant less than 1 year | 53 (37.3) | 35 (32.7) | 18 (51.4) | 0.190 |
| | Children under 5 years[4] | 35 (24.6) | 27 (25.2) | 8 (22.9) | |
| | Pregnant women | 32 (22.5) | 24 (22.4) | 8 (22.9) | |
| | Elders | 18 (12.7) | 17 (15.9) | 1 (2.9) | |
| | Guest | 1 (0.7) | 1 (0.9) | 0 (0.0) | |
| | Others (specify) | 3 (2.1) | 3 (2.8) | 0 (0.0) | |
| Physical appearance of the net | Has holes | 122 (40.9) | 107 (45.7) | 15 (23.4) | **0.010** |
| | Its burned | 20 (6.7) | 13 (5.6) | 7 (10.9) | |
| | Dead mosquitoes | 12 (4.0) | 11 (4.7) | 1 (1.6) | |
| | Look dirty / colour change | 34 (11.4) | 23 (9.8) | 11 (17.2) | |
| | Looks old | 73 (24.5) | 53 (22.6) | 20 (31.2) | |
| | Other appearance[5] | 37 (12.4) | 27 (11.5) | 10 (15.6) | |

[1] Percentage total may exceed 155 total for households owning the LLINs due to multiple responses

[2] p-value from Chi-square test or Fisher's exact test for categorical variables, two-sided; bold p-values indicate statistical significance (p<0.05).

[3] Q1, quartile 1; Q3, Quartile 3; Q3-Q1 gives the interquartile range (IQR).

[4] Not infants. Under five years but above one year.

[5] It's new, clean, enumerator didn't observe.

ago (2.43, 95% CI 0.25, 23.3; p = 0.44). There was marginal evidence of a greater odds of LLIN utilization among respondents knowledgeable of the correct cause of malaria that is, mosquito bites (AOR 3.19, 95% CI 0.77, 13.2; p = 0.11) but the estimate was not statistically significant. In the univariable model only, the results indicated that as the number of sleeping spaces

**Table 6. Frequency and washing materials of mosquito net amongst households surveyed the night prior to the survey in Mogadishu–Somalia, 2021.**

| Variable | Level | Total (%) |
|---|---|---|
| Washes the mosquito net | Yes | 134 (89.3) |
| | No | 16 (10.7) |
| Frequency of washing of LLIN | Every month | 91 (66.4) |
| | Every two months | 20 (14.6) |
| | Every three months | 1 (0.7) |
| | Every 6 months | 10 (7.3) |
| | Other[1] | 15 (10.9) |
| Washing material | Boiled water | 19 (13.9) |
| | Cold water | 13 (9.5) |
| | Soap | 27 (19.7) |
| | Omo detergent | 62 (45.3) |
| | Chlorine | 14 (10.2) |
| | Other[2] | 2 (1.5) |

[1] Every two weeks (3), every week (5)

[2] It's new.

increased, marginal odds of LLIN utilization reduced (OR 0.85, 95% CI 0.74, 0.97; p = 0.020). We also found that most of the LLINs were hung the night prior to the survey (7.9% versus 98%) and was therefore associated with greater marginal odds of utilization (p<0.001).

**Reasons for not utilizing the LLIN.** The respondents reported the "net too old/ has man holes" (41.0%) as the major reason for not using the LLINs and unable to hang (7.7%). Interestingly, 13 out of the 16 stating net too old/ has man holes and all the respondents unable to hang their nets reported that household member(s) suffered malaria in the last 12 months. The distribution is shown in Table 8 below.

## Discussion and conclusions

### Discussion

We found that ownership of the LLINs was very low. However, in the households that owned the nets, majority of the available nets were used in the night prior to the survey The reported high net-level use could be linked to the considerably moderate knowledge of preventive methods. The LLIN use we report in this study is an improvement from a low 15.7% reported by Noor et al. [22] in Rural South Central Somalia in 2008 despite considerably high *Plasmodium falciparum* infection prevalence among non-bed net users. Elsewhere, the LLIN utilization result is comparable with 84.2% reported in Madagascar [13]. The Malagasy study was after a free LLIN mass distribution and closely matches the current data on net use we report given that the majority of the nets were from mass distribution as well The low mosquito net ownership is consistent with previous reports ranking Somalia low regarding percentage of population at risk of malaria due to limited access to an LLIN, and proportion of households with adequate LLINs for all members [23].

Majority of the respondents, albeit below half for the overall population had correct knowledge of the cause of malaria. Data from a subset of those owning the nets indicated that an overwhelming number knew the correct cause of malaria hence the high levels of net utilization. Therefore, the results from overall population means that the knowledge of malaria is still low among the residents of Mogadishu districts as it negatively (barrier) influences the

**Table 7. Factors associated with reported utilization of LLIN the night prior to the survey in Mogadishu–Somalia, based on a generalized estimating equation model (n = 237).**

| Variable | Net used previous night | | Univariable Model | | | Multivariable Model | | |
|---|---|---|---|---|---|---|---|---|
| | No, N = 38[1] | Yes, N = 199[1] | OR[2] | 95% CI[3] | p-value | AOR[4] | 95% CI[3] | p-value |
| Education | | | | | | | | |
| Illiterate | 24 (63%) | 110 (55%) | — | — | | | | |
| Primary/Secondary/Tertiary | 14 (37%) | 89 (45%) | 1.50 | 0.62, 3.64 | 0.37 | | | |
| Age (years) | 28 (26, 34) | 30 (25, 40) | 1.02 | 0.99, 1.05 | 0.27 | | | |
| Marital status | | | | | | | | |
| Partnered | 25 (66%) | 161 (81%) | — | — | | — | — | |
| Unpartnered | 13 (34%) | 38 (19%) | 0.34 | 0.14, 0.82 | **0.017** | 0.60 | 0.03, 12.3 | 0.74 |
| Gender | | | | | | | | |
| Male | 4 (11%) | 30 (15%) | — | — | | | | |
| Female | 34 (89%) | 169 (85%) | 0.76 | 0.24, 2.44 | 0.65 | | | |
| Average Monthly Income (United States Dollar) | 72 (52, 100) | 90 (60, 150) | 1.00 | 1.00, 1.01 | 0.18 | | | |
| Average Monthly Income—Quartiles (United States Dollar) | | | | | | | | |
| <60 | 10 (26%) | 45 (23%) | — | — | | | | |
| 101–150 | 8 (21%) | 29 (15%) | 1.09 | 0.28, 4.22 | 0.91 | | | |
| 150+ | 1 (2.6%) | 40 (20%) | 4.66 | 0.53, 40.9 | 0.17 | | | |
| 60–100 | 19 (50%) | 85 (43%) | 1.01 | 0.39, 2.61 | 0.99 | | | |
| Distance to the nearest health facility or hospital in Kilometers (Km) | | | | | | | | |
| One Km | 19 (50%) | 101 (51%) | — | — | | | | |
| More than one Km/Don't know | 19 (50%) | 98 (49%) | 1.01 | 0.45, 2.26 | 0.98 | | | |
| Number of sleeping spaces | 4.0 (2.0, 4.75) | 2.0 (1.0, 4.0) | 0.85 | 0.74, 0.97 | **0.020** | 1.00 | 0.61, 1.64 | 0.99 |
| Sleeping places in the household | | | | | | | | |
| 2-Bed room | 30 (79%) | 157 (79%) | — | — | | | | |
| Open room | 8 (21%) | 42 (21%) | 0.95 | 0.37, 2.44 | 0.91 | | | |
| Number of people per sleeping space | 6.0 (5.0, 7.8) | 6.0 (4.0, 8.0) | 0.92 | 0.84, 1.00 | **0.047** | 0.83 | 0.74, 0.94 | **0.002** |
| Sleeping materials in household | | | | | | | | |
| Beds | 8 (21%) | 61 (31%) | — | — | | | | |
| Mats/Mattress | 30 (79%) | 138 (69%) | 0.45 | 0.13, 1.61 | 0.22 | | | |
| Interior walls of dwelling sprayed against mosquitoes | 11 (29%) | 74 (37%) | 1.71 | 0.64, 4.58 | 0.28 | 0.99 | 0.20, 4.79 | 0.99 |
| Household member(s) suffered malaria | 28 (74%) | 134 (67%) | 0.64 | 0.24, 1.71 | 0.37 | 1.24 | 0.13, 11.9 | 0.85 |
| Net hung last night | 3 (7.9%) | 196 (98%) | 724 | 137, 3,832 | **<0.001** | 1,668 | 124, 22,528 | **<0.001** |
| Months since the household obtained net | | | | | | | | |
| 12 months ago, or less than 12 months ago | 21 (55%) | 133 (67%) | — | — | | — | — | |
| More than 12 months ago | 17 (45%) | 66 (33%) | 0.60 | 0.30, 1.18 | 0.14 | 2.43 | 0.25, 23.3 | 0.44 |
| Correct knowledge of the cause of malaria (mosquito bites)[5] | 25 (66%) | 170 (85%) | 2.31 | 0.92, 5.80 | 0.076 | 3.19 | 0.77, 13.2 | 0.11 |
| Number of nets available | | | | | | | | |
| One net | 23 (61%) | 78 (39%) | — | — | | — | — | |
| Two or more nets | 15 (39%) | 121 (61%) | 2.44 | 0.96, 6.19 | 0.061 | 3.55 | 0.53, 23.5 | 0.19 |

[1] n (%); Median (IQR)

[2] OR = Odds Ratio, from a generalized estimating equation (exchangeable correlation structure and a binomial logit link)

[3] CI = Confidence Interval

[4] AOR = Adjusted odds Ratio, from generalized estimating equation with exchangeable correlation structure and a binomial logit link.

Note: bold p-value indicate statistically significant difference; '—' indicate a reference level.

Variables with p-values ≤ 0.15 threshold in the univariable model entered into the multivariable model

[5] Eating other dirty food, drinking dirty water, getting soaked with rain, cold or changing weather, dirty surrounding, witchcraft, other causes, do not know.

**Table 8. Reasons for not using the LLIN amongst household members the night prior to the survey, Mogadishu–Somalia, 2021.**

| Reasons not using net | | Total (%) | Household member(s) suffered malaria in the last 12 months | |
|---|---|---|---|---|
| | | | Yes | No |
| Net too old/Man holes | | 16 (41.0) | 13 (46.4) | 3 (27.3) |
| Unable to hang | | 3 (7.7) | 3 (10.7) | - |
| Not enough space under the net / I feel closed in | | 3 (7.7) | - | 3 (27.3) |
| Too hot under the net | | 2 (5.1) | - | 2 (18.2) |
| Don't like smell | | 1 (2.6) | - | 1 (9.1) |
| No mosquitoes | | 1 (2.6) | 1 (3.6) | - |
| Other | The house is now empty; It's dirty; cultural belief; waste; thief took it; an old tradition; found it yesterday; It's new; use electrical killer. | 13 (33.3) | 11 (39.3) | 2 (18.2) |
| Total | | 39 | 28 | 11 |

ownership of the nets as well. Importantly, sleeping under a bed net was the most cited method of protection against mosquito bites. Comparable levels of knowledge of the malaria cause and methods of prevention have been reported previously in Yaqshid district, Mogadishu-Somalia [24]. Tassew et al. [25] also reported that knowledge of malaria, including methods of prevention, is an important predictor of LLIN utilization. In this survey, most respondents identified cause of malaria as mosquito bites, however, they still cited other incorrect causes, this demonstrate that knowledge of malaria aetiology and the role of the mosquito vector is still patchy. Generally, vector control using LLINs is a tool widely implemented for malaria prevention, however, knowledge of the local community about malaria and the vector can be a barrier to the utilization of LLINs [26] especially in humanitarian contexts as in Somalia.

Lifespan/time since obtaining the LLINs was a predictor of use and non-use. Owning old LLINs appears to be a barrier to their use as results demonstrate. The WHO suggested that LLINs ought to be serviceable for not less than three years, with sufficient insecticidal activity [27]. However, previous assessments recommended a serviceable life of two years instead of three for the LLIN [28, 29]. Over the years, studies have shown variations in duration of the net's serviceable life span ranging from less than two years to more than four years [29–31]. In this current study, the median time since obtaining the nets was 12 months, and this is similar to that reported by Solomon et al. [29]. Given this, the considerably high LLIN utilization appears to be attributable to the lower timespan of the survey nets. As such, households owning considerably older nets were less likely to utilize them by their members.

Moreover, the physical integrity could also be a factor of life span of the net and its bio-efficacy. The reasons given for non-use, such as manholes and burns, accompanied by increased life span of the nets, appear to support the reported non-use in this present study. Although, the acceptable bio-efficacy for this type of net is at least two years.

Our results also demonstrated that LLIN being expensive was a major barrier to ownership even though there was no statistically significant difference in income levels of the respondents stratified by either owning a net or sleeping under a net. We found that most of the LLINs were from free mass distribution (55.7%) and market/retail shop (19.5%). This means that the local community are reliant on national or community-level mass distribution. This is

validated by the reported reason that LLINs 'are expensive' and the reported median of average income of those owning the nets (USD 90.0 [40.0]) being significantly lower than non-net-owning households. The latest LLIN distribution have been done by development partners–international agencies such as UNICEF. For instance, in 2017, UNICEF Somalia partnered with GFATM and distributed about 1.2 million LLINs through MOH and local partners [3]. These organizations have played an important role in enabling LLIN access to the people in Somalia. More efforts, are needed to sustain malaria prevention in Somalia.

We also found that marital status of the household head was an important factor of net-use/non-use. Being an unpartnered (single widowed or divorced or separated) household head/respondent was associated with not owning the nets and if available was associated with low odds of net-level LLIN use. Some female household heads reported that having lost their spouses early contributed to lack of nets in their households considering their low median average household income. All these further show that a targeted LLINs distribution in the community can considerably increase ownership and utilization of LLINs in households in Mogadishu districts and close to the campaign target of having 90% households owning at least one net [9, 32, 33].

The study also found that having a household member having suffered malaria was associated with owning a mosquito net. The LLIN non-owning and non-user households suffered malaria alike. The nets don't protect against mosquito bites/insects, nets are unavailable, and the nets are too old or have man holes as well as cultural beliefs for non-use were some of the reasons for lack of ownership and non-use, accordingly. Some of the reasons could be due the households perceiving they are less susceptible to malaria infection. Despite these reasons, LLIN ownership will not be impactful on the malaria burden if people do not sleep under them. In terms of the reasons given for, these explanations are consistent with some studies in the region, particularly Ethiopia, that reported low LLIN utilization [8, 34].

Similarly, the perceptions that malaria is not a serious threat in the study area which led to non-utilization of the nets have been reported in Tanzania [38]. However, as already been mentioned elsewhere in this study, malaria still remains a burden to the residents of Mogadishu districts and the Ministry of Health and its partners can continue playing a significant role in increasing coverage in in these areas.

This study found that children under 5 years, pregnant women and elders–in that order–were given priority in situations where there was only one LLIN in a household. Similar finding was reported in Ethiopia in which surveyed households gave priority to children under five years [35] and Tanzanian households, in which LLINs were most likely to be used by infants, young children, and women of reproductive age, but utilization was least likely among older children, older women, and adult men [36]. However, in some places, low net use by children under five years [37] have been reported.

Large household size was a barrier to bed net use. The large household size can help explain the choices the households make on who to be given priority especially when there is only net in the household. This is particularly the case since those households (about a third) owning at least an LLIN were large but the recommendation is that one mosquito net should be for every two persons at risk of malaria and increased coverage [23]. According to the Somalia Health and Demographic Survey (SHDS) [17],. The average household size for urban households is 6.6 persons per household and this can play a negative role in reducing the use of the nets especially with the low access to LLINs. Overall, this finding is comparable with that from other sub-Saharan Africa countries in which the odds of bed net use were lower in larger sized households than smaller sized households [13, 38, 39] However, a cross-sectional study, after mass distribution under a project, reported to the contrary in which case larger household size was associated use of mosquito nets but in situation [40].

## Limitations to the study

This study's data collectors were well trained to handle respondents and limit potential biases. While observation of the nets validated their presence, reporting on other study outcomes such as net use the previous night, washing frequency, average income, among others, was reliant on self-reporting which is subject to response bias. Despite that the data collected from the sample the researchers were unable to reach the total sample size calculated in this study due to logistic challenges.

## Conclusion

Ownership of LLINs was insufficient in the study area making it problematic at household-level to utilize them, therefore, universal coverage goal of ensuring that at least 90% of households own an LLIN has not been achieved in these locations, if this is not checked, this could weaken the progress made on malaria control efforts. However, the goal of ensuring at least 80% utilize at least one LLINs per two persons was achieved, largely driven by recently acquired nets. The modest use of LLIN in spite of poor ownership and access shows that there is a population-level desire to utilize them.

The study also has demonstrated that barriers of LLINutilization includes, large size of households, older LLINs, poor physical integrity of the nets, lack of knowledge of malaria cause or preventive methods, un-partnered household heads as well as misconceptions of low susceptibility of the residents to malaria infection.

The huge number of households that do not have LLINs implies that mass distribution campaigns and facility-based uninterrupted dissemination will remain relevant, especially for government and partner organizations programmes involved, however, efforts for willingness-to-pay for LLINs should also be strengthened to sustain coverage and replacements of worn-out nets. As the utilization of the nets was motivated by availability of recently acquired nets, regular sample surveys should be done to assess use of the nets in Mogadishu districts. Appropriate and purposeful health and social education on implications of frequent washing on the insecticidal activity, or the right method of washing LLINs should also be developed to safeguard the bio-efficacy of LLINs.

## Supporting information

**S1 Data. Data entry sheet.**
(XLSX)

## Acknowledgments

We are very grateful to Dr. Mohamed Ahmed Jimale. Senior researcher is Somali Research and Development Institute (SORDI) who has tremendously supported us in providing a private storage in a data server at SORDI and eventually imported the data to Excel for analysis.

## Author Contributions

**Conceptualization:** Ahmed Aweis, Fathi A. Araye, Abdifatah M. Ahmed, Osman A. Wehlie, Ali Abdirahman Osman, Isaiah Gumbe Akuku.

**Formal analysis:** Abdinur A. Salad, Abdifatah M. Ahmed, Isaiah Gumbe Akuku.

**Methodology:** Ahmed Aweis, Abdinur A. Salad, Osman A. Wehlie, Ali Abdirahman Osman, Isaiah Gumbe Akuku.

**Supervision:** Fathi A. Araye.

**Writing – original draft:** Ahmed Aweis, Abdinur A. Salad, Fathi A. Araye, Abdifatah M. Ahmed, Osman A. Wehlie, Ali Abdirahman Osman, Isaiah Gumbe Akuku.

**Writing – review & editing:** Ahmed Aweis, Abdinur A. Salad, Fathi A. Araye, Abdifatah M. Ahmed, Osman A. Wehlie, Ali Abdirahman Osman, Isaiah Gumbe Akuku.

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
