## [Decision Letter · Decision Letter 0]

15 Aug 2022

PGPH-D-22-00878

LONG-LASTING INSECTICIDE NETS (LLINs) USE AMONG HOUSEHOLD MEMBERS FOR PROTECTION AGAINST MOSQUITO BITE IN MOGADISHU DISTRICTS

Dear Mr. Ahmed Aweis,

Thank you for submitting your manuscript to PLOS Global Public Health. After careful consideration, we feel that it has merit but does not fully meet PLOS Global Public Health’s publication criteria as it currently stands. Therefore, we invite you to submit a revised version of the manuscript that addresses the points raised during the review process.

We look forward to receiving your revised manuscript.

Kind regards,

M Abdullah Yusuf

Academic Editor

Journal Requirements:

1. Please amend your ethics statement stating that the IRB specifically approved your study.

2. Please update your online Competing Interests statement. If you have no competing interests to declare, please state: “The authors have declared that no competing interests exist.”

3. We do not publish any copyright or trademark symbols that usually accompany proprietary names, eg (R), (C), or TM  (e.g. next to drug or reagent names). Please remove all instances of trademark/copyright symbols throughout the text, including ® (Olyset®) on page 19.

Additional Editor Comments (if provided):

Major correction is needed according tot he reviewers.

Reviewers' comments:

Reviewer's Responses to Questions

**Comments to the Author**

1. Does this manuscript meet PLOS Global Public Health’s publication criteria? Is the manuscript technically sound, and do the data support the conclusions? The manuscript must describe methodologically and ethically rigorous research with conclusions that are appropriately drawn based on the data presented.

Reviewer #1: Partly

Reviewer #2: Yes

2. Has the statistical analysis been performed appropriately and rigorously?

Reviewer #1: Yes

Reviewer #2: No

3. Have the authors made all data underlying the findings in their manuscript fully available (please refer to the Data Availability Statement at the start of the manuscript PDF file)?

Reviewer #1: No

Reviewer #2: No

4. Is the manuscript presented in an intelligible fashion and written in standard English?

Reviewer #1: Yes

Reviewer #2: Yes

5. Review Comments to the Author

Reviewer #1: Thank you to the authors for contributing to the understanding of LLIN ownership and use in Somalia. The authors present that though ownership in Somalia is low, net use among households that do own nets is high. This shows promising evidence that increasing net access and ownership may likely lead to increases in net use at the household level. Please see my comments below to explain the review questions and to address other minor issues:

1. I responded "partly" to the question regarding meeting publication criteria. I believe the manuscript meets the criteria, however some clarification on the ethical clearance is needed. In the manuscript, ethical considerations section, the authors state "the study was presented for review" to the institutional review board. Please provide clarification on whether the study was approved through the IRB. (As it reads, it may be interpreted that the IRB simply received the application, but does not state the decision of that review)

2. Statistical analyses. It appears the analyses have been conducted correctly. However, for replicability, more information is needed for the reader. In describing the Generalized Estimating Equations, please report what parameters were used (family, identity link function, correlation structure, and type of standard errors). For an example of GEEs used in related malaria literature, see Santos et al 2019, https://malariajournal.biomedcentral.com/articles/10.1186/s12936-019-2908-6

3. PLOS Data Policy. I responded No- however, the authors partially meet this requirement. The authors provided a Google Spreadsheet with the raw data. The only reason I state this does not meet the PLOS data policy is because the spreadsheet requires a password and only the corresponding author's contact information is provided to help provide access. Within the Data Policy statement, PLOS does not allow for only a single author to be the contact point for data access. Please review the policy and find an alternative way to allow for password access to the dataset.

Other Comments:

1. The objective of this manuscript is to measure which factors are associated with net use among households. It was found that net ownership was low (155 households own nets, or 37% of households). This reduces the sample size to 155, which likely affects the power of the GEE models to detect significant associations. Among those 155 households, net use was rather high (199 nets out of 237 total nets were used the prior night=84% use). Realizing the objective of the manuscript is to measure factors association with use- it would seem that more exploration of factors associated with net ownership would address the larger issue here. If the authors have any data on factors related to net ownership, I highly recommend incorporating this into the manuscript.

2. Section 4.4.1- The authors state that there were 2 nets per household among the households that owned nets. However, 237 total nets/155 households that own nets = 1.53 nets per household. Please correct this statement.

3. In the title, I recommend using the term "insecticidal" to remain consistent with the field and the terminology used in the abstract. This will allow for the manuscript to be found during systematic reviews and literature searches in the future.

4. Was data collected during the transmission season, or does transmission occur year-round? Please clarify the Plasmodium transmission season in the study area in relation to when data were collected (this could impact ownership and use rates)

5. On page 10, the authors state “In cases where there was only one net, the net was majorly used by children under 5 years old (37.3%) however, difference in frequency was statistically attenuated (p=0.190)." Please explain what this statement means. It appears (based on the p-value), that there was no statistical difference in age category of net use. Please clarify.

6. In the Introduction, the authors state "However, a dramatic reduction in malaria infection rates has also been shown". Please further explain this sentence. Does this refer to Somalia or a broader geographic context? In what time period, and by how much?

7. Please provide some more explanation regarding how households are acquiring nets. From the manuscript, it appears that the last distribution event was in 2013-2014, but the median net age was 12 months. Were the nets that were obtained through mass distribution older on average than those that were purchased? Or has there been a more recent distribution? Are there more distributions planned for this region?

8. Methods section: the manuscript states that the target population is all household members in the region. However, the unit of analysis is the household level, so the target population should match the unit of analysis

9. Overall, there are too many data tables. Some of the tables may be better presented as figures, and some may be better condensed into other data (particularly tables that are not directly related to the main study objectives). This is a minor comment and more personal preference (not requirement to address).

Overall, this manuscript contributes to the literature and shows important data on LLIN ownership and use. Addressing the items above will help clarify and strengthen the manuscript. Thank you to all the authors for the work and time in preparation to disseminate these results.

Reviewer #2: I would like to commend the authors for this much needed study specific to the Somalia context to understand LLIN access and use. I would recommend that the authors make the raw data accessible per the data availability section and copy edit the manuscript for grammar and flow. Kindly see below my suggestions and comments to help strengthen the academic rigor and relevance of this manuscript.

Abstract

A cross-sectional study was carried out on a sample of 409 households in Mogadishu….

Include the number of household members.

LLINs use

Define this and clarify how it was measured e.g LLIN use the previous night was assessed by asking household members to show all nets owned and confirm if the net was used.

obtained ‘12 months ago’.

Quotation marks not needed, obtained within 12 months of the survey

adjusted odds ratio [AOR] 0.227, 95% CI 0.067–0.776

Recommend 2 decimal points for the estimates.

Knowledgeable of malaria

Knowledge of malaria

Suggested revisions to other sections of the paper may have implications on the abstract.

Introduction

Paragraph 2

Mogadishu districts (Benadir region)

For readers not familiar with the Somalia context, clarify where this region is based on the classification in the first paragraph. Hypoendemic north/center or hyperendemic south.

However, a dramatic reduction in malaria infection rates has also been shown.

This sentence would benefit from more information. When was the dramatic reduction and why do the authors think this might have happened.

Because of the high numbers of morbidities and mortalities,.....

This sentence seems out of place as the authors have not provided any important information on the morbidity and mortality of Malaria in the study context. Kindly consider including it or editing this sentence

Within the framework of the Somalia National Malaria Strategic plan for 2017-2020, the

main vector control activity implemented in the country is LLIN distributions.

Clarify the other vector control activities so readers can understand the full context.

The goal of universal coverage is to ensure that at least 90% of households own an LLIN,

and at least 80% use at least one LLINs per 2 persons.

This sentence is not entirely accurate. See https://www.who.int/docs/default-source/malaria/mpac-documentation/mpac-oct2017-draft-updated-recommendations-universal-llin-coverage-session9.pdf?sfvrsn=5af603e8_2#:~:text=Ensuring%20universal%20coverage%20of%20all,populations%20at%20risk%20of%20malaria.

According to the 2013 estimates from the First Malaria Indicator Survey in Somalia.

Authors should draw attention to the lack of recent data as part of the rationale for their study.

To meet the Sustainable Development Goals (SDG) and the Roll Back Malaria targets

Authors should provide relevant examples of such goals and targets

Routine distribution remains as per the Somalia national LLIN policy guidelines focusing on pregnant women and newborns.

More information on the other channels of distribution is sorely needed. Clarify the context of EPI, ANC and private sector channels of LLIN distribution in Somalia. The study findings have direct implications on these.

There have been many LLIN distribution campaigns. National and international organizations have distributed LLINs in all districts of Mogadishu in the years 2013 and 2014.

The sentences appear contradictory and don’t imply many campaigns. Authors should specify all the distribution campaigns implemented in the study area.

Study design. This was a cross-sectional household study conducted using a stratified two-stage cluster survey design.

Authors should clarify the two stages e.g. enumeration areas and households

Sample size. According to the 2014 Somalia Malaria Indicator Survey[4] , 20.1% of household members slept under an LLIN.

The indicator above is based on household members whereas the sample size calculation was done for households. The authors should provide a justification for their approach.

The research teams used a structured questionnaire through verbal interviews and direct observation.

The authors should clarify whether they observed nets and asked who slept under them or they just simply asked the household members if they slept under a net.

Results

An overarching comment is the lack of clarity as to whether the results are for households or household members. Since the authors note that all household members were surveyed, it is unclear how Table 1 on household characteristics and Table 2 on respondents are the same N of 409.

Table 2.

The current guidance on malaria knowledge is to note the percent of respondents who correctly only state malaria as being caused by mosquito bites. The authors should note this proportion in the table.

Table 4 is hard to understand because it is unclear whether the unit of analysis is the net or the household member. The authors should explain how the different levels of information were collected in the methods section and then present net level results- number of nets observed, source, age shape, color, appearance, net used the previous night. Etc.

The row titled number of nets is confusing and unclear. It would only make sense if the data presented was at the household level.

In addition, further clarification of what the authors intend to convey is needed for the row titled “People using net if there is only one net available in the household (n=100)”. Is n=100 referring to households? Also the numbers do not add up to 100. Recommend that the authors present their information in a clear and concise way that clearly shows the rationale for the specific results.

Discussion

It is strongly recommended that the authors do not include percentages or statistics already noted in the results section

Universal coverage is no longer typically measured at the household level but better assessed at the individual or household member level

Authors should revise the discussion of correct malaria knowledge to focus on the correct and only response of mosquito bites as the cause of malaria.

“LLIN being expensive (42.0%) was a major barrier to ownership.” Authors should discuss this in relation to the most recent mass ITN distribution in the study area. This should be done for the follow up section that states “Majority of the LLINs were from free mass distribution (55.7%) and market/retail shop (19.5%). This means that the local community are reliant on national or community-level mass distribution.”

“However, knowledge of preventive methods was considerably moderate.” This sentence should be discussed in relation to ITN utilization as in the previous sentence.

“This study found that children under 5 years (61.9% households), pregnant women and elders – in that order – were given priority in situations where there was only one LLIN in a household.“ This presumes that only one person slept under a net. Can the authors verify this?

6. PLOS authors have the option to publish the peer review history of their article (what does this mean?). If published, this will include your full peer review and any attached files.

**Do you want your identity to be public for this peer review?** For information about this choice, including consent withdrawal, please see our Privacy Policy.

Reviewer #1: No

Reviewer #2: No

---

## [Decision Letter · Decision Letter 1]

15 Feb 2023

LONG-LASTING INSECTICIDE NETS (LLINs) USE AMONG HOUSEHOLD MEMBERS FOR PROTECTION AGAINST MOSQUITO BITE IN MOGADISHU DISTRICTS

PGPH-D-22-00878R1

Dear Dr Aweis,

We are pleased to inform you that your manuscript 'LONG-LASTING INSECTICIDE NETS (LLINs) USE AMONG HOUSEHOLD MEMBERS FOR PROTECTION AGAINST MOSQUITO BITE IN MOGADISHU DISTRICTS' has been provisionally accepted for publication in PLOS Global Public Health.

Best regards,

Paolo Angelo Cortesi, PhD

Academic Editor

Reviewer Comments (if any, and for reference):

Reviewer's Responses to Questions

**Comments to the Author**

1. If the authors have adequately addressed your comments raised in a previous round of review and you feel that this manuscript is now acceptable for publication, you may indicate that here to bypass the “Comments to the Author” section, enter your conflict of interest statement in the “Confidential to Editor” section, and submit your "Accept" recommendation.

Reviewer #1: All comments have been addressed

Reviewer #2: All comments have been addressed

2. Does this manuscript meet PLOS Global Public Health’s publication criteria? Is the manuscript technically sound, and do the data support the conclusions? The manuscript must describe methodologically and ethically rigorous research with conclusions that are appropriately drawn based on the data presented.

Reviewer #1: Yes

Reviewer #2: Yes

3. Has the statistical analysis been performed appropriately and rigorously?

Reviewer #1: Yes

Reviewer #2: Yes

4. Have the authors made all data underlying the findings in their manuscript fully available (please refer to the Data Availability Statement at the start of the manuscript PDF file)?

Reviewer #1: Yes

Reviewer #2: Yes

5. Is the manuscript presented in an intelligible fashion and written in standard English?

Reviewer #1: Yes

Reviewer #2: Yes

6. Review Comments to the Author

Reviewer #1: (No Response)

Reviewer #2: (No Response)

7. PLOS authors have the option to publish the peer review history of their article (what does this mean?). If published, this will include your full peer review and any attached files.

**Do you want your identity to be public for this peer review?** For information about this choice, including consent withdrawal, please see our Privacy Policy.

Reviewer #1: No

Reviewer #2: No
